# Racial and Ethnic Comparisons in Satisfaction with Services Provided by the Special Supplemental Nutrition Program for Women, Infants, and Children in California

**DOI:** 10.3390/nu15020447

**Published:** 2023-01-14

**Authors:** Alana M. Chaney, Lorrene D. Ritchie, Shannon E. Whaley, Marisa M. Tsai, Hallie R. Randel-Schreiber, Catherine E. Yepez, Susan Sabatier, Adrian Young, Martha Meza, Lauren E. Au

**Affiliations:** 1Department of Nutrition, University of California Davis, Davis, CA 95616, USA; 2Nutrition Policy Institute, Division of Agriculture and Natural Resources, University of California, Oakland, CA 94607, USA; 3Public Health Foundation Enterprises WIC, 12781 Schabarum Avenue, Irwindale, CA 91706, USA; 4Communicable Disease Prevention Unit, San Francisco Department of Public Health, 101 Grove St, Rm 406, San Francisco, CA 94102, USA; 5WIC Division, California Department of Public Health, 3901 Lennane Drive, Sacramento, CA 95834, USA

**Keywords:** WIC, children, satisfaction, nutrition education

## Abstract

Understanding satisfaction of nutrition education and other services provided in the Special Supplemental Nutrition Program for Women, Infants and Children (WIC) is needed to ensure the program is responsive to the needs of diverse populations. This study examined the variation of WIC participants’ perceptions and satisfaction with WIC nutrition education and services by race, ethnicity, and language preference. Phone surveys were conducted in 2019 with California WIC families with children aged 1–4 years. While most participants (86%) preferred one-on-one nutrition education, online/mobile apps were also favored (69%). The majority (89%) found nutrition education equally important to receiving the WIC food package. Racial/ethnic groups differed in which WIC service they primarily valued as 20% of non-Hispanic White people rated the food package as more important than nutrition education compared to 5% of Spanish- and 6% of English-speaking Hispanic people, respectively. More Spanish (91%) and English-speaking Hispanic people (87%) than non-Hispanic white (79%) or Black people (74%) changed a behavior because of something they learned at WIC (*p* < 0.001). Spanish-speaking Hispanic people (90%) had the highest satisfaction with WIC nutrition education. Preferential differences among participants suggest that providing flexible options may improve program satisfaction and emphasizes the need for future studies to examine WIC services by race and ethnicity.

## 1. Introduction

The Special Supplemental Nutrition Program for Women, Infants and Children (WIC) provides nutrition education and supplemental healthy foods to 6.4 million nutritionally at-risk low-income women, infants, and young children in the United States [1]. Administered by the United States Department of Agriculture (USDA) Food and Nutrition Service (FNS), the program has grown rapidly since it was first established in 1972 [2]. Children aged 1–4 years are the largest group of WIC beneficiaries representing over half of all WIC participants [1]. Participation in WIC has been shown to improve household food purchases [3], food security [4,5] child dietary intakes [6,7,8,9] and weight status [10,11], highlighting the need to ensure at-risk families participate in WIC during early childhood years [12].

WIC is the only USDA FNS program federally required to provide nutrition education, which is intended to improve nutritional status, prevent nutrition-related problems [13], and improve health status [12,14,15]. Local WIC agencies provide various delivery modes of education, such as in-person one-on-one sessions, group sessions, or technology-based resources (e.g., online education, videos/DVDs). The USDA considers in-person contact with WIC staff to be the optimal education delivery mode. However, recent technological changes to WIC practices due to the COVID-19 pandemic and transition from paper vouchers to electronic benefit transfer (EBT) cards [16] have increased the use of technology-based resources in WIC. In addition, a national WIC study determined that local agencies often adapted their education services according to the race and ethnic composition of participants suggesting a need to examine nutrition education preferences by race and ethnicity [17].

Understanding WIC participant satisfaction with nutrition education ensures its most unique service is accommodating to individual needs and meets current public health challenges, a mission of WIC’s Revitalizing Quality Nutrition Services (RQNS) program [18]. Therefore, the objective of this study is to examine WIC participants’ perceptions and satisfaction with WIC’s nutrition education by race, ethnicity, and language preference. In addition, this study serves as a baseline for WIC nutrition education utilization by race and ethnicity prior to the COVID-19 pandemic.

## 2. Materials and Methods

A multi-disciplinary team from the California WIC program, Public Health Foundation Enterprises (PHFE) WIC Program, and the Nutrition Policy Institute at the University of California Division of Agriculture and Natural Resources conducted the study. The California Health and Human Services Agency’s Committee for the Protection of Human Subjects approved the study.

### 2.1. Survey Respondents

A sample of nearly 10,000 children, ages 1–4 years with an active WIC certification record as of August 2018 was randomly selected from state administrative records to reach a goal of 3000 survey respondents. This goal was selected assuming a 5% type 1 error rate and 90% power to detect a 2–3% difference in the proportion of respondents reporting a behavior before and after the main study’s EBT rollout in California [19]. In October 2018, families with a preferred language of English or Spanish per original WIC enrollment were sent postcards alerting them to an upcoming call to complete a survey. Multiple call attempts were then made to each household at varying times of day and different days of the week. After obtaining oral consent from each child’s caregiver (hereafter referred to as respondent), screener questions were asked to verify study eligibility. Respondents were excluded from completing the survey if they were <18 years of age, unable to complete the survey in English or Spanish, no longer living in California, not currently caring for a child of 1–4 years, if they had only a foster child in WIC or a child with a condition preventing them from eating most WIC foods. Respondents who were previously certified in WIC during the study selection period but no longer enrolled in WIC at the time of the phone call were eligible to complete the survey. Respondents who had more than one child in WIC between the ages of 1 and 4 were asked to complete the survey for the child whose birthday was closest to the survey date.

### 2.2. Survey Development and Administration

Survey questions were adapted from previous surveys [9,20,21], pilot tested with 8 English-speaking and 11 Spanish-speaking participants and revised as needed. From late January to early June 2019, surveys were administered by phone in English or Spanish using a computer-assisted telephone interview (CATI) system by Davis Research, LLC, an independent survey firm. All interviewers were trained and certified in interviewing techniques specific for this study. Surveys required approximately 30–35 min to complete (average of 32 min in English, 33 min in Spanish). Each respondent was mailed a USD 10 gift card after survey completion.

Demographics and variables of interest were collected from two sources, WIC administrative records and survey data. WIC administrative records were used to determine child sex and the following sociodemographic information was collected from respondents in the survey: race and ethnicity, primary language preference (English or Spanish), highest level of education completed, current employment status, number of young children in the household (<5 years old), current participation in SNAP and Medicaid, total duration and current participation in WIC, and child age. Questions with multiple response options were asked to evaluate respondent satisfaction with WIC nutrition education and reasons for remaining on the program, and household food insecurity was determined using the USDA 2-item validated screener [20].

### 2.3. Data Analysis

The following respondent racial and ethnic and language groups were examined: non-Hispanic White (hereafter referred to as White), English-speaking Hispanic, Spanish-speaking Hispanic, Non-Hispanic Black (hereafter referred to as Black), Asian, and Other (which included respondents identifying as two or more races, Pacific Islander, American Indian or other races). English and Spanish-speaking Hispanic respondents were examined separately based on prior studies showing differences in satisfaction with WIC based on primary language [19,22]. Racial and ethnic group frequencies were compared using chi-square tests. If expected cell counts were below 5, Fisher’s exact test was used to adjust for multiple comparisons, statistical significance by race–ethnicity and primary language was defined using a Bonferroni approach as *p* < 0.003 (*p* = 0.05/15). Data are presented for the total sample and stratified by racial–ethnic groups. Data were analyzed using SAS software (version 9.4, SAS Institute Inc., Cary, NC, USA).

## 3. Results

### 3.1. Respondent Characteristics

A total of 2,997 respondents completed the survey with a response rate of 30%. Study children were 2.5 years old on average, and half were female (Table 1). Participants that did not report race and ethnicity (*n* = 60) or had insufficient data (*n* = 4) were excluded from analyses, bringing the analytical sample to 2933 (Figure 1). Of the survey respondents, 49.3% were Spanish-speaking Hispanic, 25.7% English-speaking Hispanic, 12.8% White, 5.1% Other, 4.1% Black, and 3.0% Asian. Two-thirds of survey respondents had a minimum of a high school education; among Spanish-speaking Hispanic respondents, over half had not completed high school. One in six White and Black respondents and over one in four Asian respondents were college graduates. Overall, 42.1% of respondents were working for pay, ranging from 35.8% of Spanish-speaking Hispanic to 50.9% of Black respondents. Most children under study (93.9%) remained in WIC at the time of the survey, 86.6% of households participated in Medicaid, and 37.8% participated in SNAP. Participation in WIC for the household averaged nearly 6 years. Spanish-speaking Hispanic households had the longest time of participation in WIC, averaging nearly 7 years, while Asian households had the shortest time at 4 years. Half of the families lived in food insecure households.

### 3.2. WIC Nutrition Education

Most survey respondents reported receiving one-on-one nutrition education from WIC staff (91.3%), followed by watching a video or DVD (74.7%), attending a group session (62.5%), or taking an online class (57.5%) (Table 2). Two-way texting which is conducted behind a safe firewall and allows WIC staff to tailor individual messages [23] was reported by less than one-third of respondents (30.0%). Differences in modes of WIC nutrition education except for one-on-one were significant (*p* = <0.001) for all groups. Compared to other groups, more Spanish-speaking Hispanic respondents attended group sessions (70.7%) and fewer used online education (43.5%). Asian respondents (42.7%) reported the most use of two-way texting compared to White respondents (22.5%) who reported the least. Videos/DVDs were reported most by Spanish-speaking Hispanic respondents (77.5%) and least used by White respondents (67.9%).

In terms of the preferred modes of receiving WIC nutrition education in the future, in-person one-on-one mode was selected by 86.4% of survey respondents, followed by video/DVD (75.9%), online (68.9%), group session (62.3%), and two-way texting (55.4%) modes (Table 2). Over two-thirds (68.6%) of survey respondents reported that they would like to use a WIC mobile app, a mode of delivering nutrition education not available in California WIC at the time of the survey. Significant differences existed between groups for all modes of WIC nutrition education preferred in the future except for two-way texting. Compared to the other groups, more Spanish-speaking Hispanic respondents would like to receive one-on-one (89.1%), video/DVD (83.6%), and group session (73.5%) education, and fewer would like to receive education online (59.9%) or via mobile app (61.7%). 

Overall, most respondents were very satisfied with WIC nutrition education (83.6%) and customer service (87.2%). Levels of satisfaction with nutrition education were highest among Spanish-speaking Hispanic respondents (90.1%) and lowest among respondents in the Other racial–ethnic group (77.8%) (Table 2). Respondents also reported that they changed nutrition behavior because of something learned from WIC (86.8% overall). Changing a behavior because of WIC nutrition education was reported by more Spanish-speaking Hispanic respondents (90.8%) than White (79.0%), Black (73.7%) and respondents in the Other racial–ethnic group (80.9%). Across-group comparisons on satisfaction with WIC customer service did not differ.

### 3.3. Value of WIC Participation

When asked about reasons for continuing to participate in WIC, most survey respondents reported the fruits and vegetables provided in the child WIC food package (92.5%), followed by information from WIC staff (87.0%), support from WIC staff (86.0%), and classes and group sessions (65.7%) (Table 3). The most common reason (fruits and vegetables) and least common reason (classes and group sessions) reported were similar for all racial and ethnic groups, yet differences across groups were significant (*p* < 0.001). Spanish-speaking Hispanic respondents selected more reasons for participating in WIC while White, Asian, and Black respondents selected fewer reasons. When asked whether WIC food or education benefits were more important or both were equally important, 89.6% of the overall sample reported that both were equally important. Spanish-speaking Hispanic respondents (93.7%) were more likely than other groups to report that foods and nutrition education were equally important. Compared to the other racial and ethnic groups, more respondents in the White (20.4%) and Other (12.4%) groups rated WIC foods as more important than WIC nutrition education or both.

## 4. Discussion

Study findings in 2019, before the COVID-19 pandemic, suggest that caregivers with young children in WIC were highly satisfied with the nutrition education and customer service provided in California. Overall, Spanish-speaking Hispanic respondents valued WIC nutrition education more than other racial and ethnic groups did. This was also determined in a study focused on WIC food package changes which reported a higher percentage of Spanish-speaking participants being satisfied compared to English speakers [19]. In a recent study, more Spanish speakers also reported being satisfied than English speakers with WIC nutrition education regardless of delivery mode [22]. All racial and ethnic groups of survey respondents reported one-on-one education with WIC staff as the most preferred mode of receiving education in the future which aligned with the most common way they currently received education from WIC before the pandemic. Survey respondents also reported that they would like to receive more online education. Many respondents currently used online education, and even more reported that they would like to use this format in the future, which has also been supported through studies conducted during the pandemic [24,25,26]. While both one-on-one and group sessions were in person, a substantial number of participants preferred one-on-one sessions in the future. This difference may be due to the personalization of one-on-one sessions where participants can often choose the topic, whereas group session topics are usually determined based on the attending participants (e.g., classes on nursing for pregnant or breastfeeding parents) [17]. Previous studies have also reported that most WIC sites offered more group sessions on breastfeeding and fewer sessions on topics that may be of more interest to caregivers of young children such as child feeding practices and fruits and vegetables [17].

In the current study, Spanish-speaking Hispanic respondents were the only group to choose videos/DVDs as their secondary mode of education delivery, whereas other groups chose either online or mobile app. When comparing online and WIC mobile app education, all groups preferred online education except Spanish-speaking Hispanic and Black respondents. This is supported by previous research that showed low-income Hispanic and Black individuals often underutilize the internet as they are more likely to be mobile-only users; however, differences between English- and Spanish-speaking Hispanic individuals were not reported [27]. It has also been shown that preference for online delivery of WIC nutrition education increases with exposure to it—particularly among those who have not used online education previously, such as some Spanish speakers [22]. This has been explored during COVID-19 WIC adaptations where a multi-state survey determined that the majority of participants received and highly rated nutrition education by phone [28]. Determining whether WIC participant’s preferences for nutrition education delivery have evolved since COVID-19 should be assessed in future studies.

Pre-pandemic in-person nutrition education, especially one-on-one education, offered the benefit of a personalized experience allowing the participant to establish their own nutrition goals which has been shown to be more effective [17]. However, due to COVID-19, WIC sites have recently incorporated more hybrid forms of education using technology-based delivery resources, such as online education [13]. Results from this study suggest that most participants prefer to receive a mixture of in-person and technology-based delivery options, ultimately desiring more flexibility to meet their WIC education requirements. One reason WIC participants may prefer a hybrid model of nutrition education may involve issues with accessing the WIC clinic. A 2016 study determined that those who were eligible for WIC faced barriers such as transportation, finding childcare, and the inability to leave work which prevented many from participating [21]. Similarly, a Food Research and Action Center (FRAC) report determined that eligible families not participating in WIC experienced high transportation costs to WIC clinics and lost wages due to taking time off from work for appointments [29]. Because accessing WIC appointments may become increasingly challenging as infants grow into mobile toddlers, nutrition education delivery preferences should also be explored for other WIC participants such as pregnant women and caregivers of infants.

A national study on WIC nutrition education determined that WIC staff reported difficulty in delivering nutrition education to participants that are more interested in solely receiving cash benefits from WIC [17]. However, most respondents in the present study reported that WIC nutrition education was as important as cash benefits. The most frequently reported reason for continuing WIC for all racial and ethnic groups, aside from Asian individuals, was the cash value benefit (CVB) provided for the purchase of fruits and vegetables. At the time of this study, the CVB provided USD 9 a month for children which has since been temporarily increased due to COVID-19 and the American Recovery bill. Aside from fruits and vegetables, most respondents also reported information and support from WIC staff as top reasons for participating in WIC. As COVID-19 has modified several WIC services, continually assessing participants’ preferences and satisfaction can help inform policy leaders to establish effective programs.

Language barriers have been identified in prior studies as one challenge with WIC participation [21,30]. For example, a language barrier may not only impact communication about WIC eligibility, but it can also affect participants’ comfort level with staff, understanding nutrition education, appointment scheduling and even attendance [30]. In our study sample, Spanish-speaking Hispanic individuals tended to be more satisfied with WIC and WIC nutrition education delivery compared to the other, majority English-speaking respondents. This is consistent with previous WIC studies where Spanish speakers in California were determined to be more satisfied with nutrition classes [22] and food package options [31]. Notably, we did not include respondents who were unable to speak English or Spanish. Future studies should examine experiences of participants who speak other languages, such as Asian dialects, as previous studies suggest that translations are underutilized in these communities [21].

A strength of this study is the fact that California serves one in seven of all WIC participants nationally and accommodates some of the most racially and ethnically diverse populations. Therefore, understanding perceptions of California WIC’s nutrition education program is crucial to understanding the program nationally. Few studies exploring the impacts of WIC services have been able to determine a direct benefit of nutrition education on the nutritional behavior of participants. The study also has limitations, as it was conducted pre-pandemic. While participant preferences in modality of education may have changed, this study emphasizes the importance of future studies examining WIC nutrition education by race and ethnicity and language, as differences do exist. Another limitation is that all survey responses were self-reported to an interviewer, which may have biased responses given that respondents were informed that the survey was being conducted by WIC. In addition, results are not representative of California as a whole, as Spanish speakers were overrepresented in the total sample.

## 5. Conclusions

Overall, WIC services were highly rated by California WIC participants who has children in the program, regardless of race and ethnicity and language spoken. These findings suggest that offering flexible options for nutrition education and more fruits and vegetables in the food package may improve long-term retention of children in WIC. As California WIC’s response to the COVID-19 pandemic has shifted most of WIC operations to remote services, this study highlights participants’ value of in-person interactions supporting a future ‘hybrid’ model. Future studies are warranted to evaluate newly adapted WIC nutrition educational practices on participant satisfaction and future preferences.

## Figures and Tables

**Figure 1 nutrients-15-00447-f001:**
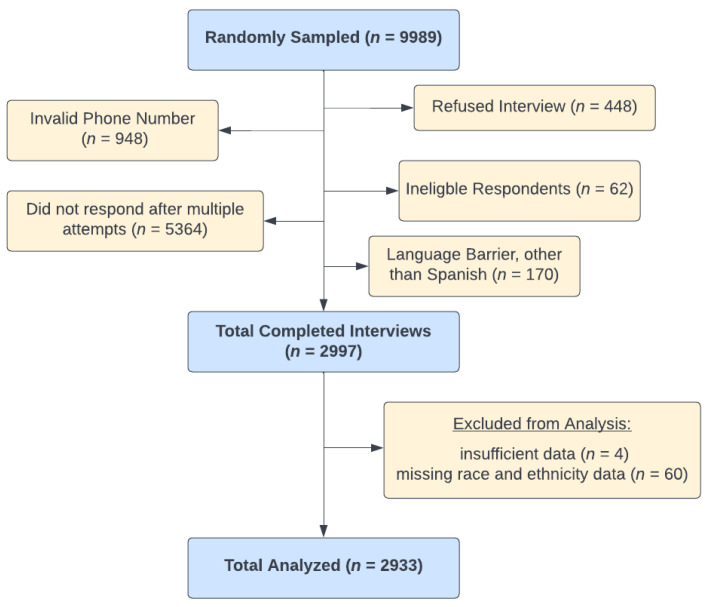
Flow diagram of caregivers with children aged 1–4 years selected from California WIC state administrative records.

**Table 1 nutrients-15-00447-t001:** Characteristics of 2019 California WIC Survey Sample by Respondent Race/Ethnicity ^1^.

	Total (*n* = 2993)	Race/Ethnicity ^2^					
		Non-Hispanic White (*n* = 374)	Hispanic English-Speaking (*n* = 753)	Hispanic Spanish-Speaking (*n* = 1445)	Non-Hispanic Black (*n* = 119)	Asian (*n* = 89)	Other ^3^ (*n* = 153)
Age of child under study (mean years, SD)	2.5 (1.1)	2.4 (1.1)	2.5 (1.0)	2.6 (1.1)	2.4 (1.0)	2.4 (0.9)	2.4 (1.0)
Sex of child under study (*n*, % female)	1439 (49.1)	179 (47.9)	359 (47.7)	705 (48.8)	68 (57.1)	44 (49.4)	84 (54.9)
Child under study currently in WIC (*n*, %)	2753 (93.9)	343 (91.7)	694 (92.2)	1380 (95.5)	112 (94.1)	80 (89.9)	144 (94.1)
Number of months household participated in WIC (mean, SD)	70.7 (52.4)	61.6 (47.7)	59.3 (46.6)	81.9 (54.7)	74.4 (60.8)	50.9 (45.4)	51.8 (39.3)
Number of young children (<5 y) in household (mean, SD)	1.2 (0.5)	1.3 (0.5)	1.3 (0.5)	1.2 (0.4)	1.3 (0.5)	1.3 (0.5)	1.3 (0.5)
Highest level of education of respondent (*n*, %)							
8th grade or less	468 (16.4)	19 (5.1)	3 (0.4)	437 (30.6)	0 (0)	1 (1.2)	8 (5.2)
Some high school	5494 (17.0)	32 (8.6)	79 (10.5)	356 (24.9)	8 (6.7)	6 (7.0)	13 (8.5)
High school graduate	946 (32.5)	119 (31.8)	270 (35.9)	455 (31.8)	35 (29.4)	20 (23.3)	47 (30.7)
Some college or trade school	735 (25.2)	141 (37.7)	329 (43.8)	105 (7.3)	57 (47.9)	35 (40.7)	68 (44.4)
College graduate or more	271 (9.3)	63 (16.8)	71 (9.4)	77 (5.4)	19 (16.0)	24 (27.9)	17 (11.1)
Respondent working for pay ^4^ (*n*, %)	1229 (42.1)	176 (47.2)	375 (49.8)	515 (35.8)	60 (50.9)	36 (41.4)	67 (43.8)
Household receives SNAP ^5^ (*n*, %)	1103 (37.8)	147 (39.3)	280 (37.3)	513 (35.7)	71 (60.2)	24 (27.0)	68 (44.7)
Household receives Medicaid (*n*, %)	2515 (86.6)	309 (82.6)	640 (85.5)	1250 (88.0)	109 (91.6)	74 (83.2)	133 (86.9)
Household experienced food insecurity in prior 12 months (*n*, %)	1459 (49.8)	200 (53.5)	378 (50.2)	682 (47.3)	60 (50.4)	47 (52.8)	92 (60.1)

^1^ WIC refers to Special Supplemental Nutrition Program for Women, Infants, and Children. Totals may not add up to 100% due to rounding. ^2^ Race/ethnicity is reflective of study respondent and caregiver of the study child. ^3^ Other includes two or more races, Pacific Islander, American Indian or other. ^4^ Working for pay includes full-time or part-time employment. ^5^ SNAP refers to Supplemental Nutrition Assistance Program.

**Table 2 nutrients-15-00447-t002:** Satisfaction with WIC Nutrition Education as Reported by Caregivers of Young Children on WIC in California ^1^.

	Total (*n* = 2933)	Race/Ethnicity ^2^						
		Non-Hispanic White (*n* = 374)	Hispanic English-Speaking (*n* = 753)	Hispanic Spanish-Speaking (*n* = 1445)	Non-Hispanic Black (*n* = 119)	Asian (*n* = 89)	Other ^3^ (*n* = 153)	*p* Value ^4^
	(*n*, %)							
Modes of WIC nutrition education currently received ^5^								
One-on-one in person	2678 (91.3)	334 (89.3)	685 (91.0)	1334 (92.3)	110 (92.4)	80 (89.9)	135 (88.2)	0.3
Video/DVD	2192 (74.7)	254 (67.9) ^a^	559 (74.2) ^ab^	1120 (77.5) ^b^	84 (70.6) ^ab^	65 (73.0) ^ab^	110 (71.9) ^ab^	0.004
Group session in person	1832 (62.6)	189 (50.5) ^ac^	434 (57.6) ^a^	1021 (70.7) ^b^	50 (42.0) ^c^	49 (55.1) ^ac^	89 (58.2) ^ac^	<0.001
Online	1682 (57.5)	254 (67.9) ^a^	551 (73.2) ^a^	628 (43.5) ^b^	79 (66.4) ^a^	67 (75.3) ^a^	103 (67.3) ^a^	<0.001
Two-way texting	875 (29.8)	84 (22.5) ^a^	245 (32.5) ^b^	421 (29.1) ^ab^	31 (26.1) ^ab^	38 (42.7) ^b^	56 (36.6) ^b^	<0.001
Modes of WIC nutrition education preferred in future ^5^								
One-on-one in person	2535 (86.4)	309 (82.6) ^a^	635 (84.3) ^a^	1287 (89.1) ^b^	96 (80.7) ^ab^	76 (85.4) ^ab^	132 (86.3) ^ab^	0.002
Video/DVD	2227 (75.9)	239 (63.9) ^a^	538 (71.5) ^a^	1208 (83.6) ^b^	76 (63.9) ^a^	67 (75.3) ^ab^	99 (64.7) ^a^	<0.001
Online	2021 (68.9)	277 (74.1) ^a^	610 (81.0) ^a^	866 (59.9) ^b^	83 (69.8) ^ab^	73 (82.0) ^a^	112 (73.2) ^a^	<0.001
WIC mobile app	2017 (68.8)	273 (73.0) ^a^	588 (78.1) ^a^	891 (61.7) ^b^	90 (75.6) ^a^	63 (70.8) ^ab^	112 (73.2) ^ab^	<0.001
Group session in person	1826 (62.3)	177 (47.3) ^ac^	412 (54.7) ^a^	1062 (73.5) ^b^	47 (39.5) ^c^	53 (59.6) ^abc^	75 (49.0) ^ac^	<0.001
Two-way texting	1624 (55.4)	197 (52.7)	409 (54.3)	828 (57.3)	55 (46.2)	51 (57.3)	84 (54.9)	0.17
Very satisfied with WIC nutrition education ^6^	2528 (86.3)	295 (79.1) ^a^	646 (85.8) ^ab^	1300 (90.1) ^b^	94 (79.0) ^a^	74 (83.2) ^ab^	119 (77.8) ^a^	<0.001
Very satisfied with WIC customer service ^6^	2555 (87.2)	321 (85.8)	637 (84.6)	1285 (89.1)	99 (83.9)	82 (92.1)	131 (85.6)	0.07
Changed behavior because of WIC nutrition education	2537 (86.8)	294 (79.0) ^c^	652 (86.7) ^ab^	1308 (90.8) ^a^	87 (73.7) ^c^	73 (83.0) ^abc^	123 (80.9) ^bc^	<0.001

^1^ WIC refers to Special Supplemental Nutrition Program for Women, Infants, and Children. Totals may not add up to 100% due to rounding. ^2^ Race/ethnicity is reflective of study respondent and caregiver of the study child. ^3^ Other includes two or more races, Pacific Islander, American Indian or other. ^4^ Statistical significance assessed using chi-square test; values sharing a common superscript (a–d) are not significantly different from each other using a Bonferroni approach at a 5% procedure-wise error rate. ^5^ Participants asked to choose all that applied with answer options presented in a random order. ^6^ Rated on a Likert scale (not at all satisfied, not too satisfied, somewhat satisfied, very satisfied); results presented for respondents indicating very satisfied.

**Table 3 nutrients-15-00447-t003:** Value of WIC Participation as Reported by Caregivers of Young Children on WIC in California ^1^.

	Total (*n* = 2933)	Race/Ethnicity ^2^						
		Non-Hispanic White (*n* = 374)	Hispanic English-Speaking (*n* = 753)	Hispanic Spanish-Speaking (*n* = 1445)	Non-Hispanic Black (*n* = 119)	Asian (*n* = 89)	Other ^3^ (*n* = 153)	*p* Value ^4^
	(*n*, %)							
Reasons for participating in WIC								
Fruits and vegetables	2541 (92.5)	306 (89.2) ^a^	616 (88.8) ^a^	1335 (97.0) ^b^	92 (82.1) ^a^	68 (85.0) ^a^	124 (86.7) ^a^	<0.001
Information from WIC staff	2383 (87.0)	246 (71.9) ^a^	554 (80.5) ^b^	1323 (96.0) ^c^	78 (69.6) ^ab^	69 (86.3) ^ab^	113 (80.1) ^ab^	<0.001
Support from WIC staff	2357 (86.0)	259 (76.0) ^a^	538 (77.9) ^a^	1316 (95.6) ^b^	68 (60.7) ^c^	65 (81.3) ^a^	111 (79.3) ^a^	<0.001
Classes and group sessions	1790 (65.7)	161 (47.2) ^a^	378 (54.5) ^ac^	1089 (79.6) ^b^	52 (46.4) ^ac^	49 (61.3) ^ac^	61 (43.0) ^ac^	<0.001
Relative importance of WIC nutrition education versus WIC foods								
Equally important	2624 (89.4)	289 (77.5) ^a^	672 (89.4) ^b^	1354 (93.7) ^c^	102 (85.7) ^abc^	77 (87.5) ^bc^	130 (85.0) ^ab^	<0.001
Food is more important	232 (7.9)	76 (20.4)	45 (6.0)	72 (5.0)	14 (11.8)	6 (6.8)	19 (12.4)	
Education is more important	74 (2.6)	8 (2.1)	35 (4.7)	19 (1.3)	3 (2.5)	5 (5.7)	4 (2.6)	

^1^ WIC refers to Special Supplemental Nutrition Program for Women, Infants, and Children. Totals may not add up to 100% due to rounding. ^2^ Race/ethnicity is reflective of study respondent and caregiver of the study child. ^3^ Other includes two or more races, Pacific Islander, American Indian or other. ^4^ Statistical significance between race and ethnicities for Reasons for participating in WIC assessed using chi-square test; statistical significance for Relative importance of WIC nutrition education versus WIC foods assessed using Fisher’s exact test; values sharing a common superscript (a–d) are not significantly different from each other using a Bonferroni approach at a 5% procedure-wise error rate.

## Data Availability

Data will be available upon request.

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
