# Peer review of "Racial and Ethnic Comparisons in Satisfaction with Services Provided by the Special Supplemental Nutrition Program for Women, Infants, and Children in California"

_nutrients, 2023, doi:10.3390/nu15020447_

Round 1
Reviewer 1 Report
Detailed comments:
1. Please reformulate the title not to use the abbreviation. The same comment applies to keywords.
2. I suggest changing the abbreviation from WIC into e.g. SSNP (the Special Supplemental Nutrition Program) + WIC (Women, Infants and Children) to underline the importance of the nutrition programme (not only vulnerable group).
3. Lines 38-40: please provide more details for the programme (who is responsible for it? Where and when is/was it conducted? Etc.)
4. Lines 73-75: please add a reference (or a comment) for these assumptions.
5. Please add a graph showing the number of identified respondents, inclusion and exclusion criteria, the final number of participants.
6. Please clearly define who is a respondent: a caregiver or a child herself/himself as it is mixed in different part of the manuscript.
7. Line 152: what is ‘Two-way texting’? Please expand in the text and in the table.
8. Lines 224-226 & 262-264 &280-281: please provide more details.
Reviewer 2 Report
The manuscript by Chaney et al., sought to investigate satisfaction among families of various Californian ethnic groups who were provided with various nutrition services and education by a special program targeting the at-risk population. Among the main findings, the large majority of Spanish-speaking Hispanics valued the nutrition education provided by the program.
Overall the study is well-conceived and the manuscript clearly written. However, data presentation and especially Tables 1-3, in my opinion, are rather sloppy and need a thorough review.
Major criticism
Methods. Did the authors check whether the percentage of respondents excluded due to the criteria reported (page 2, lines 81-86) was comparable between the various ethnic groups? Especially for those excluded due to their inability to continue the phone interview in English or Spanish?
Why wasn't a analysis based on sex performed, given that this variable had also been collected?
Results. On page 3, lines 127-129, the percentages of the various ethnic groups appear to have been calculated on a total of 2933 and not 2993, in fact in the footnotes of the tables it is stated that ethnicity was missing in 60 cases. However, this information should be provided in paragraph 3.1 and not only in the tables.
I have noticed a strange lack of accuracy in preparing the tables. In the footnotes, the authors state that the totals may differ from the expected 100% due to rounding, but in many cases, the difference is not limited to decimals. In the heading, the sum of the subgroups representing the different ethnic groups does not make 2993 but 2933 due to missing data in 60 cases. Then, the percentages should be calculated on the valid data, i.e. 2933, not on 2993. This creates big discrepancies in my opinion. For example in table 3 on page 6, “fruits and vegetables” had to be 2591/2933=88.3 and not 92.5%. These discrepancies may not change the substance of the results but are nevertheless an indication of poor data presentation.
Minor remarks
Why was the mean age of the study child reported in Table 1 and not that of the respondent?
Round 2
Reviewer 2 Report
The authors responded adequately to most of my requests, and when this was not possible, they justified it satisfactorily. I have nothing else to ask.